# Constrained non-negative networks for a more explainable and interpretable classification

**Valentine Wargnier-Dauchelle**[1]                    VALENTINE.WARGNIER@CREATIS.INSA-LYON.FR

**Thomas Grenier**[1]                    THOMAS.GRENIER@CREATIS.INSA-LYON.FR

**Françoise Durand-Dubief**[1,2]                    FRANCOISE.DURAND-DUBIEF@CHU-LYON.FR

**François Cotton**[1,3]                    FRANCOIS.COTTON@CHU-LYON.FR

**Michaël Sdika**[1]                    MICHAEL.SDIKA@CREATIS.INSA-LYON.FR

[1] *INSA Lyon, Universite Claude Bernard Lyon 1, CNRS, Inserm, CREATIS UMR 5220, U1294, Lyon, France*

[2] *Service de Neurologie, Hôpital Neurologique, Hospices Civils de Lyon, Bron, France*

[3] *Service de Radiologie, Centre Hospitalier Lyon-Sud, Hospices Civils de Lyon, Pierre-Bénite, France*

**Editors:** Accepted at MIDL 2024

## Abstract

Interpretability and explainability of deep networks are essential for medical image analysis. Easily explainable networks with intrinsic properties and decisions based on radiological signs and not spurious confounders are highly desirable. The guaranteed monotonic relation between the input and the output of monotonic networks could be used to design such intrinsically explainable networks, but they are rarely used for images: state-of-the-art architectures are often very shallow due to convergence problems. Identifying the critical importance of weights initialization, we propose a recipe to transform any architecture into a trainable monotonic network. By using the monotonic property, adding a calibration and constraining the training in an unsupervised way, we propose a network more explainable with human-readable counterfactual examples but also more interpretable with a decision more based on the radiological signs of the pathology. Especially, we outperform state-of-the-art methods for weakly supervised anomaly detection.

**Keywords:** Monotonic network, Constrained learning, Interpretability, Anomaly detection

## 1. Introduction

Deep learning has proved its efficiency for medical image analysis but its lack of transparency compromises its use in critical areas such as medicine. A more human-readable explanation and interpretable deep networks with, ideally, guaranteed properties is therefore welcomed. For a healthy vs pathological classification, we also expect the decision to be based on the radiological signs of the pathology. Attributions (Selvaraju et al., 2017) or counterfactual examples (Wachter et al., 2017) methods can be used to explain the network decision indicating the more important areas for the decision in the input image. Nevertheless, they can be difficult to interpret as they indicate both positive and negative contributions in the decision. In terms of interpretability, adding constraints when training a classifier can help focus the decision on relevant factors (Ross et al., 2017; Wargnier-Dauchelle et al., 2023) and perform weakly-supervised segmentation. In terms of explainability, monotonic networks benefit from interesting intrinsic properties but they are rarely used for images as they are harder to train and small architectures with fewer capacities

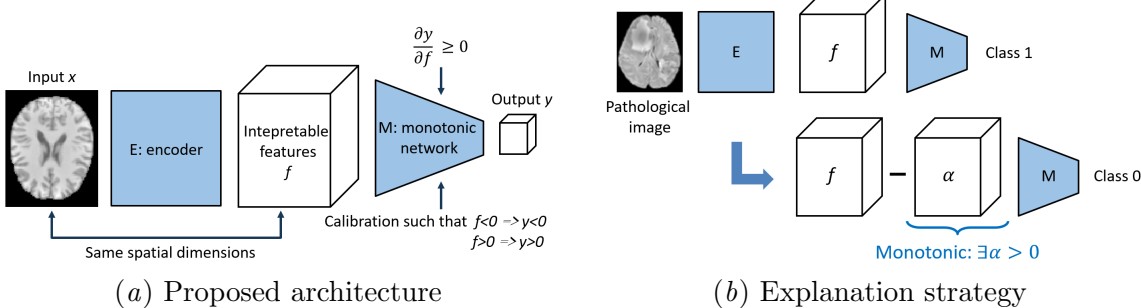

$(a)$ Proposed architecture    $(b)$ Explanation strategy

Figure 1: (a) The image $x$ is passed to the encoder E which outputs features $f$ with the same spatial dimension as $x$. They are passed to a calibrated monotonic classifier M which outputs the classification logits $y$. (b) To explain the network decision, we search the counterfactual difference $\alpha$ such that $f - \alpha$ is classified as healthy (class 0) from the features $f$ of a pathological (class 1) sample.

are often used to achieve convergence (Daniels and Velikova, 2010; Liu et al., 2020). In this work, we: 1/ identify weight initialization as a key issue for monotonic networks convergence and propose an initialization that solves this problem, 2/ leverage monotonic network properties to design a constrained and calibrated framework to improve both the explainability and the interpretability of a healthy vs pathological images classifier, 3/ outperform state-of-the-art for weakly-supervised anomaly segmentation.

## 2. Method

We design the architecture described in Figure $1(a)$ for an explainable and interpretable healthy vs pathological images classification. It is composed of an encoder E that outputs the interpretable features $f$, followed by a calibrated monotonic network M that computes the logits $y$ of the binary classification. In our case, $M$ is monotonically increasing w.r. to its input: $\frac{\partial M}{\partial f} \geq 0$. The interpretable features space thus benefits from several intrinsic explicability properties: it is ordered, with a known bound between healthy and pathological samples and counterfactual examples are more readable as we can find a *positive* $\alpha$ such that $M(f - \alpha)$ is "healthy" for a pathological feature tensor $f$ (see Figure $1(b)$). By thresholding the counterfactual difference $\alpha$, we can segment the pathology in a weakly supervised way.

To guarantee that the network M is monotonic, the weights are parameterized to be non-negative, normalization layers are removed and activation functions are convex increasing on half of the channels and concave increasing on the other half. We showed that removing biases enforces the following calibration: if $f < 0$ (resp. $f > 0$) then $y < 0$ (resp $y > 0$).

We can prove that using non-negative weights increases the correlation between the features: state-of-the-art random initialisation methods are then unsuitable for large non-negative networks and when the network depth is too large, the training is impossible. So, we proposed a new weights initialization, rescaling each linear layer, one after another, by its standard deviation, that maintains a unit-variance of all the features.

Finally, we constrain the network during training for a more interpretable classification without adding more annotation than the image's class. For that, we trained our network with 4 losses: one for the classification, one to constrain the interpretable features $f$ to be

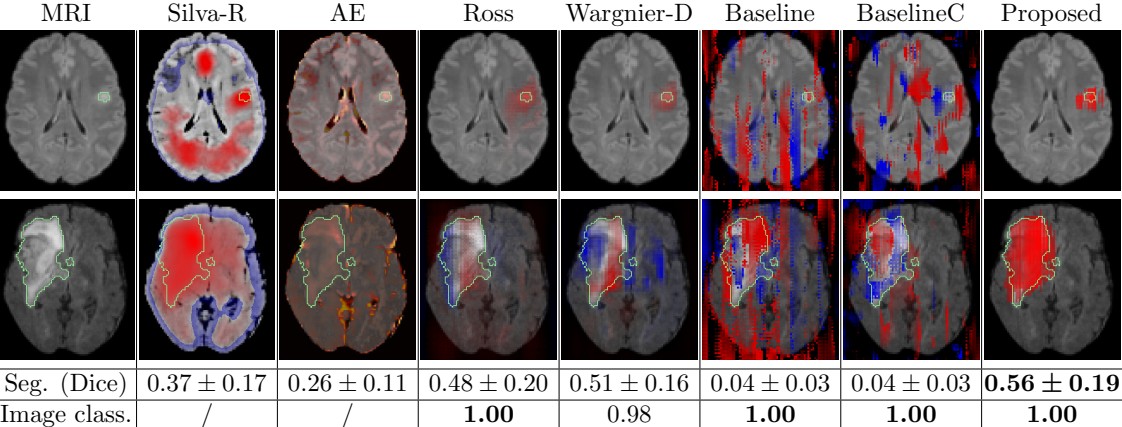

| | MRI | Silva-R | AE | Ross | Wargnier-D | Baseline | BaselineC | Proposed |
|---|---|---|---|---|---|---|---|---|
| Seg. (Dice) | | $0.37 \pm 0.17$ | $0.26 \pm 0.11$ | $0.48 \pm 0.20$ | $0.51 \pm 0.16$ | $0.04 \pm 0.03$ | $0.04 \pm 0.03$ | $\mathbf{0.56 \pm 0.19}$ |
| Image class. | | / | / | **1.00** | 0.98 | **1.00** | **1.00** | **1.00** |

Figure 2: Segmentation map and metrics (computed on the whole test database). Ground truth is in green. Blue represents healthy relevance and red pathological relevance for classification attribution methods. High attributions are in red for Silva-Rodríguez. For counterfactual examples, negative values of $\alpha$ are in blue, positive values are in red. The reconstruction error for AE ranges from black to yellow.

negative for healthy images, one to enforce similar distribution of the negative interpretables features $f$ for the both classes and a regularization of $\nabla M$ for healthy images.

## 3. Experiments and Results

Experiments are conducted using IBC (Pinho et al., 2018), kirby21 (Landman et al., 2011) and MPI (Babayan et al., 2019) healthy brain MRI datasets and Brats2020 glioma dataset (Menze et al., 2014). E is a C8(7)-C16(3) network and M is a C64(4)-C128(4)-C256(4)-C512(4)-C1(4) network, where $Cn(k)$ is a 3D convolutional layer with $n$ filters and a kernel size of $k$. We compared our weakly-supervised segmentation method to anomaly detection methods (AE (Baur et al., 2018), Silva-R (Silva-Rodríguez et al., 2021)) and other classification based methods (Ross (Ross et al., 2017), Wargnier-D (Wargnier-Dauchelle et al., 2023)). We also used two baselines: the same architecture as our proposition but non-monotonic and unconstrained (Baseline), and a non-monotonic architecture but constrained as proposed (BaselineC). The results are shown in Figure 2.

Visually, the decision of the proposed network seems more focused on the tumor as this is the area to be modified to change the decision from pathological to healthy. The counterfactual difference is also more readable than the baselines as it is positive. In terms of segmentation, we outperform the state-of-the-art methods with at least a 5-point improvement of the Dice. The image classification is perfect.

## 4. Conclusion

In this work, we propose a more interpretable network for healthy vs pathological classification with a decision more based on radiological signs of the pathology outperforming state-of-the-art methods for weakly-supervised segmentation. With the proposed calibrated non-negative network, we also benefit from intrinsic properties that increase the explainability. In particular, the counterfactual examples are more human-readable.

## Acknowledgments

This work was supported by the LABEX PRIMES (ANR-11-LABX-0063, ANR-11-IDEX-0007) in a lab member of France Life Imaging network (ANR-11-INBS-0006). This work was performed using HPC resources from GENCI-IDRIS (AD011012544/AD011012589). Finally, this work was partly funded by APIDIFF: "Projet Emergence", CNRS-INS2I.

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
