# OpenReview forum: "Constrained non-negative networks for a more explainable and interpretable classification"
_MIDL.io/2024/Short_Papers — MIDL 2024 Short Papers_

### Official Review · Reviewer_oUaX · 2024-04-23

**Confidence:** 4
**Final Rating:** 4

**Review:**

Summary
-------

The paper proposes an inherently interpretable network for image classification and anomaly detection that is composed of a feature extractor and an interpretable monotonic classification network.

Strengths
---------

- The proposed method based on monotonic networks should be an interesting addition to the field of inherently interpretable models of medical image analysis. I believe our community would be interested in this work and its followup.
- The authors did well in packing a lot of relevant information in the short paper.

Weaknesses
----------

- Could the authors describe the network architecture used for feature extraction? The receptive field size of the feature extraction network should influence the inherent interpretability of the attribution maps.

- The baseline methods (ablations of the proposed method without enforcing monotonicity) show some strange streaking artefacts. Can the authors explain these?

---

### Decision · Program_Chairs · 2024-04-26

Accept